# Identification of core genes involved in the response of *Apocynum venetum* to salt stress based on transcriptome sequencing and WGCNA

Xi Zhen[1,2], Xuyang Liu[3], Xiaoming Zhang[1], Shujie Luo[4], Wencheng Wang[5], Tao Wan[1] *

**1** Key Laboratory of Grassland Resources of Ministry of Education, College of Grassland, Resources and Environment, Inner Mongolia Agricultural University, Inner Mongolia Hohhot, China, **2** Inner Mongolia Weather Modification Center, Inner Mongolia Hohhot, China, **3** Inner Mongolia Climate Center, Hohhot, China, **4** College of Plant Protection, Yangzhou University, Yangzhou, China, **5** Jining Medical University, Jining, China

* wantao425@sohu.com

**Data Availability Statement:** Datasets are available at the NCBI project GSE229255.

**Funding:** The author(s) received no specific funding for this work.

## Abstract

*Apocynum venetum* L. belongs to the Apocynaceae family and is a plant that is highly resistant to stress. It is important in the fields of ecology, feeding, industry and medicine. The molecular mechanism underlying salt tolerance has not been elucidated. In this study, RNA-seq based transcriptome sequencing of *A. venetum* leaves after 0, 2, 6, 12, 24 and 48 h of treatment with 300 mM NaCl was performed. We conducted a comprehensive analysis of the transcriptome expression profiles of *A. venetum* under salt stress using the WGCNA method and identified red, black, and brown as the core modules regulating the salt tolerance of *A. venetum*. A co-expression regulatory network was constructed to identify the core genes in the module according to the correlations between genes. The genes TRINITY_DN102_c0_g1 (serine carboxypeptidase), TRINITY_DN3073_c0_g1 (SOS signaling pathway) and TRINITY_DN6732_c0_g1 (heat shock transcription factor) in the red module were determined to be the core genes. Two core genes in the black module, TRINITY_DN9926_c0_g1 and TRINITY_DN7962_c0_g1, are pioneer candidate salt tolerance-associated genes in *A. venetum*. The genes in the brown module were mainly enriched in two pathways, namely photosynthesis and osmotic balance. Among them, the TRINITY_DN6321_c0_g2 and TRINITY_DN244_c0_g1 genes encode aquaporin, which is helpful for maintaining the cell water balance and plays a protective role in defending *A. venetum* under abiotic stress. Our findings contribute to the identification of core genes involved in the response of *A. venetum* to salt stress.

## Introduction

The complex regulatory mechanisms of plant morphology, physiology, and genetic traits are the overall manifestation of adaptation to salinization [1]. Under salt stress plant growth and

**Competing interests:** The authors have declared that no competing interests exist.

development may be inhibited, and death is even possible. Salt ions can cause three types of stress on plants: osmotic stress, ion stress, and secondary damage. The presence of a large amount of $Na^+$ and $Cl^-$ could cause ion toxicity and nutrient deficiency. Excessive accumulation of Na+ and Cl- ions in plant tissues causes serious ion imbalance in cells, affecting the absorption of nutrients such as K, N, P, Ca, Mg and Fe, resulting in physiological and metabolic disorders, nutrient deficiency, growth retardation, and even death of plants [2, 3]. Ion toxicity and osmotic stress can also lead to the production of many reactive oxygen species (ROS), leading to oxidative stress [2–5]. Salt stress can also lead to a reduction in photosynthesis, resulting in a substantial decrease in plant yield [6]. In the face of ionic stress, osmotic stress, and oxidative stress, plants must activate a series of physiological and biochemical regulatory mechanisms to achieve ion homeostasis, osmotic balance, and detoxification to maintain normal plant growth and development. Therefore, the adaptation of plants to salt stress is a comprehensive regulatory process that requires the synergy of various physiological, biochemical, and metabolic processes.

*Apocynum venetum* is an important wild germplasm resource in China. Because the fibre quality of *A. venetum* stalks is excellent, *A. venetum* is known as the "king of wild fibres" [7]. The medicinal value of *A. venetum* has also been widely studied. *A. venetum* leaf extract has the effects of treating hypertension, lowering cholesterol, and protecting the heart [8–10]. *A. venetum* is also a plant with high stress resistance; is widely distributed in saline-alkali deserts, desert borders, riverbanks, alluvial plains, and the Gobi desert between 35˚ and 45˚N in China; and has an important ecological value [11, 12]. *A. venetum* grows in harsh environments in most parts of China; however, few studies have focused on its mechanisms of adaptation to the environment. As for the adaptation mechanism of Apocynum apocynum under salt stress, some scholars have tried to clarify the salt tolerance mechanism of Apocynum apocynum at the molecular level. Studies on the transcriptome level have shown that flavonoids are related to the excellent salt tolerance of *A. venetum*. Due to the accumulation of flavonoids, transgenic *Arabidopsis thaliana* showed higher salt tolerance than wild-type (WT) *A. thaliana*. To test this hypothesis, four flavonoid synthesis pathway genes, *AvF3H*, *AvF3′H*, *AvFLS*, and *AvF3GT*, were overexpressed in *A. thaliana*. The germination of transgenic plants were significantly better than the that of WT plants at 3 days under 100 mM NaCl stress. In addition, the roots of transgenic plants were longer than those of the WT plants under the condition of 100 mM salt stress. Based on these results, it is speculated that transgenic plants are more salt tolerant than wild-type plants, which may be related to the accumulation of total flavonoids [13]. The *A. venetum* DEAD-box helicase gene (*AvDH1*), stably inherited in the cotton genome, significantly improved the salt tolerance of transgenic cotton lines. Digital gene expression (DGE) analysis was performed on transgenic AvDH1 cotton and acceptor lu613. The 161 putative RNA helicase genes had been identified in the genome of the diploid cotton species *Gossypium raimondii*. These helicases had been classified into three subfamilies, which include the *DEAD-box*, *DEAH-box*, and *DExD/H-box* gene families. The *AvDH1* protein shows a sequence identity of 74% to *GrDExD/H4* and *GrDExD/H30*. Syntenic analysis revealed that *Gr DExD/H4* and *Gr DExD/H30* had syntenic relationship. Transcriptome sequencing data demonstrated that *Gr DExD/H4* and *Gr DExD/H30* were expressed at high levels in all three samplesand may play important roles in various processes of *Gossypium raimondii* [14]. Overexpression of the *A. venetum* flavanol synthase/flavanone 3-hydroxylase gene (*AvFLS*) can increase the flavonoid content and photosynthetic rate of *A. thaliana* and reduce the malondialdehyde content, thereby significantly improving the salt tolerance of transgenic *A. thaliana* [15]. Although there are some research results inthe medicinal components, genetic diversity, and salt tolerance of A. venetum, it is very necessary to conduct in-depth research in the molecular mechanism of the salt tolerance of *A. venetum*.

This study analysed the transcriptome dataset of *A. venetum* under salt stress at six time points and identified core salt stress-responsive genes. Quantitative proteomics of A. venetum using TMT technology under salt stress, the core protein obtained from differential protein network interactions at different stress times were mainly ribosomal proteins, indicating that the ribosomal proteins played an important role of A. venetum in response to salt stress. Furthermore, by applying the weighted gene co-expression network analysis (WGCNA), this study identified three core modules regulating salt stress tolerance in *A. venetum*. In this study, the molecular resources related to the response of *A. venetum* to salt stress were expanded, and a theoretical basis is provided for the exploration and utilization of *A. venetum*, a natural stress-resistant germplasm.

## Results

### Statistical analysis of raw transcriptome sequencing data

In this study, 18 samples were sequenced by the Illumina HiSeq platform. A total of 117.98 G of clean data were obtained, and the effective data volume of each sample was more than 6.07 G. Raw reads ranged from 43264458 to 51285720. The filtered clean reads ranged from 42975126 to 51285576, the total base number was 6150108311 to 7314153091, and the total base number after quality control was 6069861180 to 7301018739. The distribution of Q30 bases was above 94.33%, and the average GC content was 43.6% (Table 1), indicating that the sequencing quality of the samples in this experiment was high. All indexes of the original transcriptome sequencing data met the data requirements.

### Transcriptome de novo assembly

De novo splicing of samples connects partially overlapping reads into a longer sequence, and after continuous extension, the reads are spliced into a transcript. The longest one is selected as the unigene, and a final set of unigene is obtained by eliminating redundancy. The total

**Table 1. Transcriptome sequencing data statistics of *Apocynum venetum*.**

| Sample | Raw reads | Raw bases | Clean reads | Error rate (%) | Q20 (%) | Q30 (%) | GC content (%) |
|--------|-----------|-----------|-------------|----------------|---------|---------|----------------|
| T0h1 | 43264458 | 6150108311 | 43264428 | 0.0267 | 98.19 | 94.66 | 43.77 |
| T0h2 | 49807320 | 7203242988 | 49807248 | 0.0275 | 98.08 | 94.33 | 43.4 |
| T0h3 | 46978818 | 6669905930 | 46978776 | 0.0258 | 98.36 | 95.14 | 44.24 |
| T2h1 | 45283380 | 6428435036 | 45283330 | 0.0258 | 98.38 | 95.18 | 43.43 |
| T2h2 | 46669586 | 6654138239 | 46669522 | 0.0263 | 98.3 | 94.96 | 43.86 |
| T2h3 | 44552540 | 6276833907 | 44552496 | 0.0258 | 98.39 | 95.21 | 43.51 |
| T6h1 | 45674740 | 6508789290 | 45674694 | 0.0261 | 98.34 | 95.06 | 43.61 |
| T6h2 | 43830356 | 6207178620 | 43830102 | 0.026 | 98.34 | 95.08 | 43.54 |
| T6h3 | 46752440 | 6639296169 | 46752338 | 0.026 | 98.33 | 95.06 | 43.49 |
| T12h1 | 43637300 | 6194561351 | 43637170 | 0.0266 | 98.26 | 94.81 | 43.67 |
| T12h2 | 44819692 | 6343170682 | 44819658 | 0.0262 | 98.32 | 95.02 | 43.5 |
| T12h3 | 44812670 | 6336813870 | 44812614 | 0.0259 | 98.37 | 95.15 | 43.64 |
| T24h1 | 44385414 | 6290590549 | 44385228 | 0.0262 | 98.3 | 94.97 | 43.51 |
| T24h2 | 42975166 | 6081550203 | 42975126 | 0.026 | 98.35 | 95.1 | 43.66 |
| T24h3 | 48713020 | 6846431525 | 48712902 | 0.0237 | 98.78 | 96.33 | 42.83 |
| T48h1 | 50795468 | 7115424245 | 50795384 | 0.0238 | 98.73 | 96.23 | 43.55 |
| T48h2 | 49281342 | 6937559559 | 49281232 | 0.0235 | 98.8 | 96.43 | 43.51 |
| T48h3 | 51285720 | 7314153091 | 51285576 | 0.0243 | 98.65 | 95.99 | 43.68 |

**Table 2. Unigene assembly statistics.**

| Type | Total number | Total base | Largest length (bp) | Smallest length (bp) | Average length (bp) | N50 length (bp) | Fragment mapped percent (%) | GC percent (%) |
|------|------|------|------|------|------|------|------|------|
| Unigene | 104283 | 85503165 | 17274 | 201 | 819.91 | 1691 | 68.323 | 42.73 |
| Transcript | 157220 | 193066889 | 17274 | 201 | 1228 | 2494 | 83.128 | 41.05 |

number of Unigene segments was 104,283, the longest segment length was 17274, the minimum segment length was 201, and the average length was 819.91. The total number of transcripts was 157220, the longest length was 17274, the minimum length was 201, and the average length was 1228. Other related splicing statistics of unigenes and transcripts (N50, Fragment mapped percent, GC percent) are shown in the following table (Table 2). The resulting length of the unigenes was distributed in the maximum length direction of 200–500 bp and 501–1000 bp (Fig 1).

## Analysis of unigene function annotation results

We compared 104,283 unigene sequences with public databases GO, KEGG, COG, NR, Swiss-Prot and Pfam (Table 3). The number of unigenes compared to each database accounted for 38.91%, 32.78%, 52.74%, 49%, 40.9% and 40.17% of the total genes, respectively. The number of unigene expressed was similar to the total number of unigene, and the percentage of unigene expressed was similar to the database, which were 38.98%, 32.81%, 52.8%, 49.08%, 40.97% and 40.23%, respectively.

## Statistics of differentially expressed genes

To explore the differences in gene transcription levels of *A.venetum* at different stress time points, 5 differential genes of comparison groups T2_vs_T0, T6_vs_T0, T12_vs_T0,

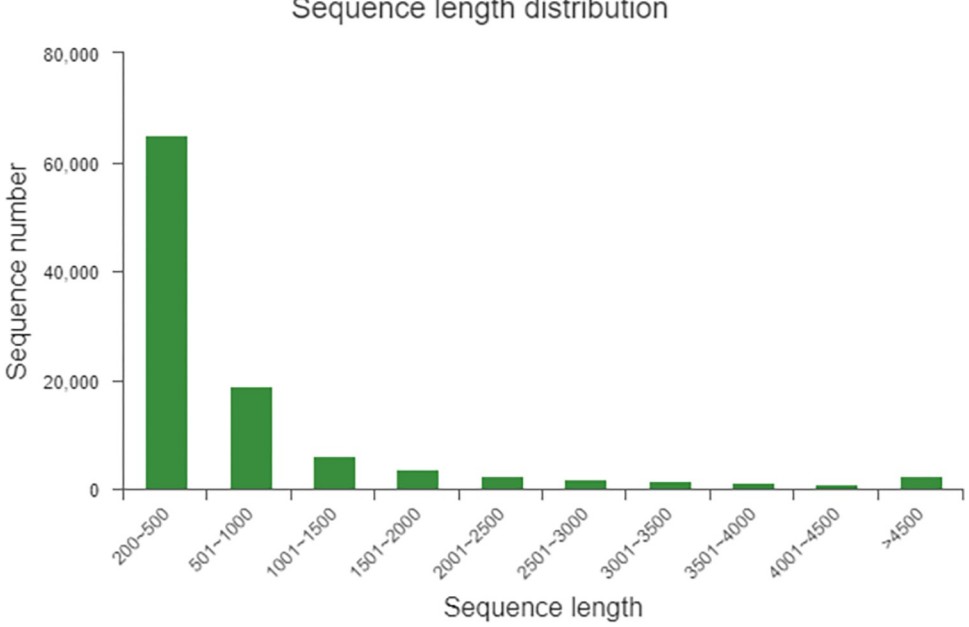

**Fig 1. Length distribution of unigenes.**

**Table 3. Unigene annotation data of *Apocynum venetum*.**

| Anno Database | Exp_Unigene number(percent) | All_Unigene number(percent) |
|---|---|---|
| GO | 40520(38.98%) | 40578(38.91%) |
| KEGG | 34105(32.81%) | 34183(32.78%) |
| COG | 54885(52.8%) | 54998(52.74%) |
| NR | 51018(49.08%) | 51100(49%) |
| Swiss-Prot | 42589(40.97%) | 42655(40.9%) |
| Pfam | 41820(40.23%) | 41888(40.17%) |
| Total_anno | 62492(60.12%) | 62626(60.05%) |
| Total | 103950(1) | 104283(1) |

T24_vs_T0 and T48_vs_T0 were selected when the p value was less than 0.05 and the difference multiple was greater than 2. A total of 11914 differentially expressed genes were obtained. A total of 373 differentially expressed genes were identified in the T2_vs_T0 group, 112 of which were upregulated and 261 of which were downregulated. There were 1903 differentially expressed genes in the T6_vs_T0 group; 941 were upregulated, and 962 were downregulated. There were 4733 differentially expressed genes in the T12_vs_T0 group, with 2196 upregulated and 2537 downregulated. A total of 5929 differentially expressed genes were identified in the T24_vs_T0 group, with 3292 upregulated and 2637 downregulated. There were 8659 differentially expressed genes in the T48_vs_T0 group, including 4770 upregulated and 3889 downregulated genes (Fig 2A). The number of differentially expressed genes increased gradually with the extension of stress time, and the number of upregulated genes was higher than that of downregulated genes.

Venn analysis showed that there were 61 differentially expressed genes in groups T2_vs_T0, T6_vs_T0, T12_vs_T0, T24_vs_T0 and T48_vs_T0 (Fig 2B). Hierarchical cluster analysis of common differentially expressed genes at these five time points showed that T6_vs_T0 and T12_vs_T0 were grouped into one group, T24_vs_T0 and T48_vs_T0 were grouped into one group, and T2_vs_T0 was grouped into one group alone. The expression of most genes was upregulated at 2 h, 6 h and 12 h in the early stage of stress, but downregulated at the late stage of stress. A few other genes were downregulated in the early stage of stress (2 h, 6 h, 12 h), and gradually upregulated with the extension of stress time (Fig 2C).

## GO classification and functional enrichment analysis of differentially expressed genes

The selected differentially expressed genes were compared and classified in the GO database and were mainly divided into three categories: cellular component (CC), molecular function (MF) and biological process (BP). All differentially expressed genes were annotated with GO function. The differentially expressed genes were mainly involved in catalytic activity (GO:0003824), binding (GO:0005488), cellular part (GO:0044464), cell membrane part (GO:0044425), organelle (GO:0043226), cellular process (GO:0009987) and metabolic process (GO:00) 08152) (Fig 3).

GO enrichment analysis of differentially expressed genes at different stress times showed that the T2_vs_T0 group was significantly enriched in hemicellulose metabolism, REDOX enzyme activity, polysaccharide metabolism and other pathways. The functional pathways of differentially expressed genes in the T6_vs_T0 group were DNA methylation maintenance, regulation of other biological processes, and ribonuclease III activity. In the T12_vs_T0 group, there were significantly enriched pathways such as reaction to chitin, polysaccharide

A

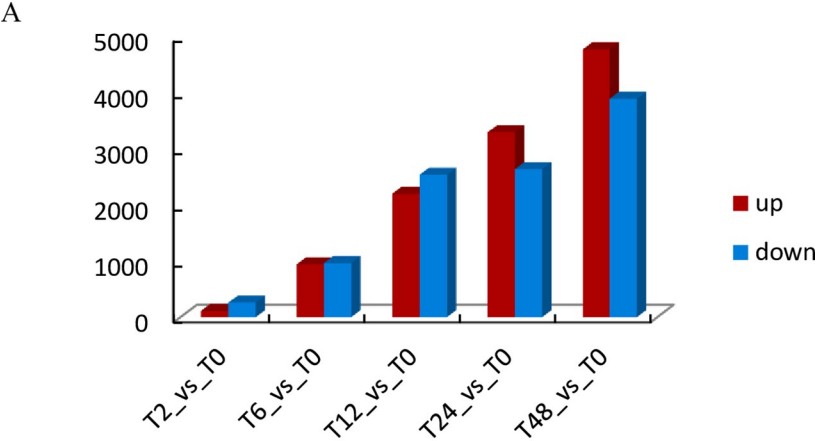

B

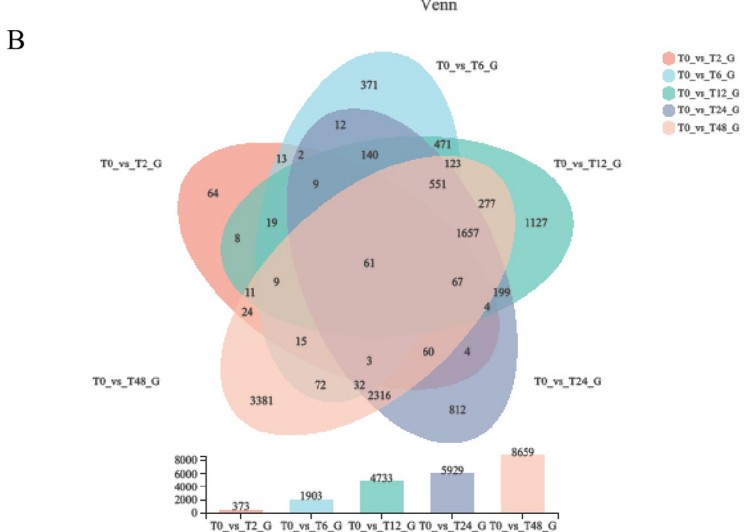

C

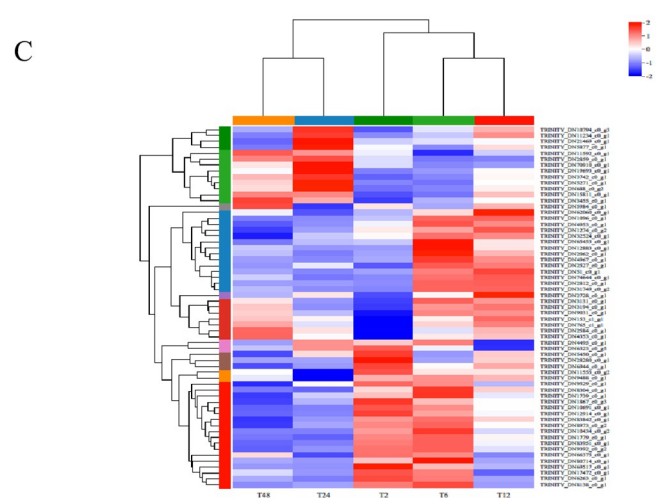

**Fig 2. DEGs of *Apocynum venetum* under different times of salt stress compared with the control.** A Number of DEGs in different comparison groups. B Venn diagram of DEGs. C Heatmap of differentially expressed genes.

metabolism, carbohydrate binding, transcriptional regulation activity, and serine/threonine protein kinase activity. The T24_vs_T0 group was significantly enriched in pectin metabolism, pectin catabolism, cell wall modification, galacturonic acid metabolism, hormone-mediated signaling pathways and other pathways. Cell wall modification, photosynthesis, pectin metabolism and photosystem I pathways were significantly enriched in the T48_vs_T0 group (Fig 4). At the five stress time points, differentially expressed genes were significantly enriched in REDOX enzyme activity, serine/threonine protein kinase activity and photosystem I pathways, which were closely related to plant response to salt stress.

## KEGG enrichment analysis of differentially expressed genes

We further analysed the KEGG pathway enrichment of differentially expressed genes. The results showed that in the T2_vs_T0 group, 373 differentially expressed genes were significantly enriched in 32 pathways, among which zeatin biosynthesis, plant hormone signal

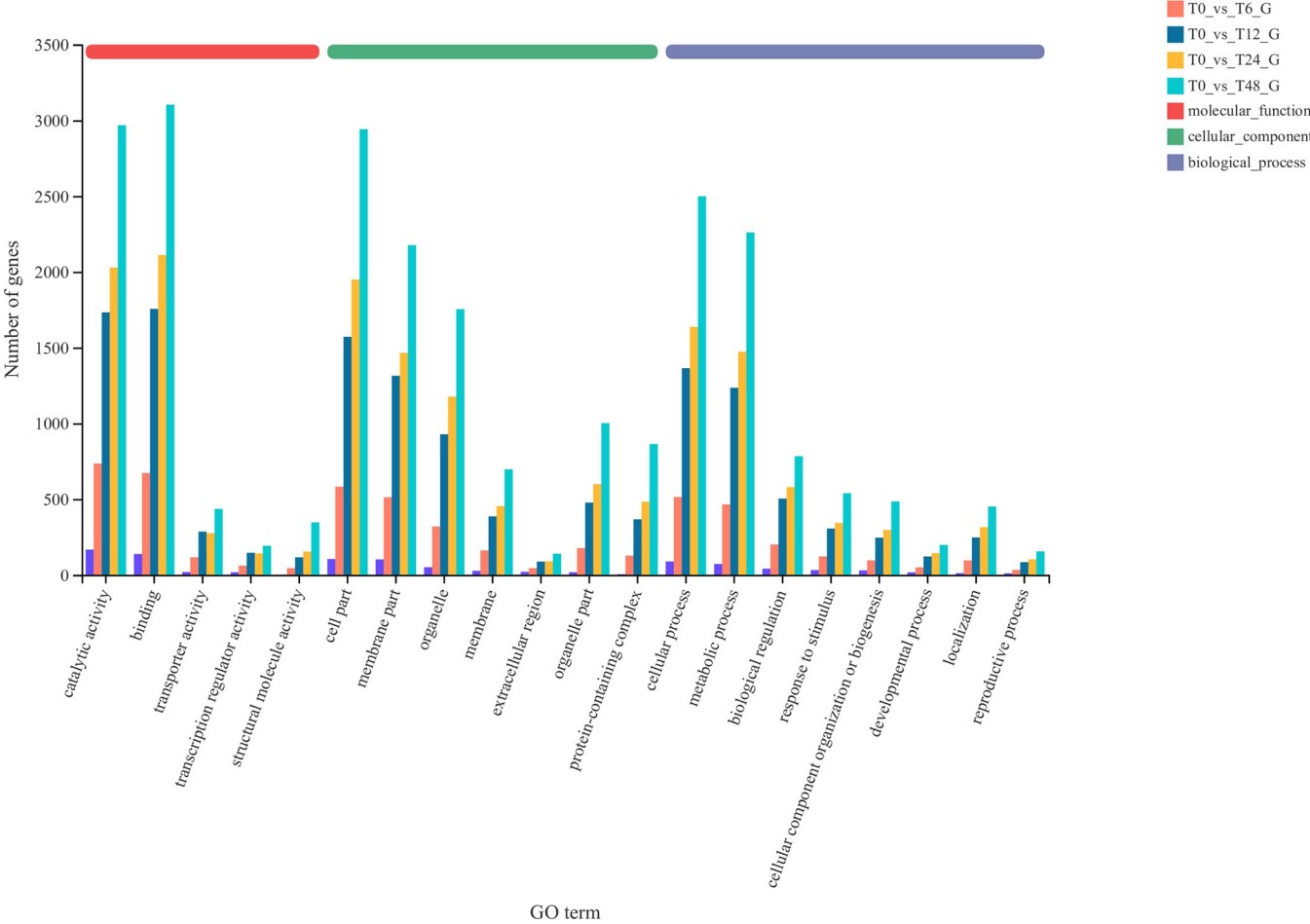

**Fig 3. GO function analysis of DEGs.**

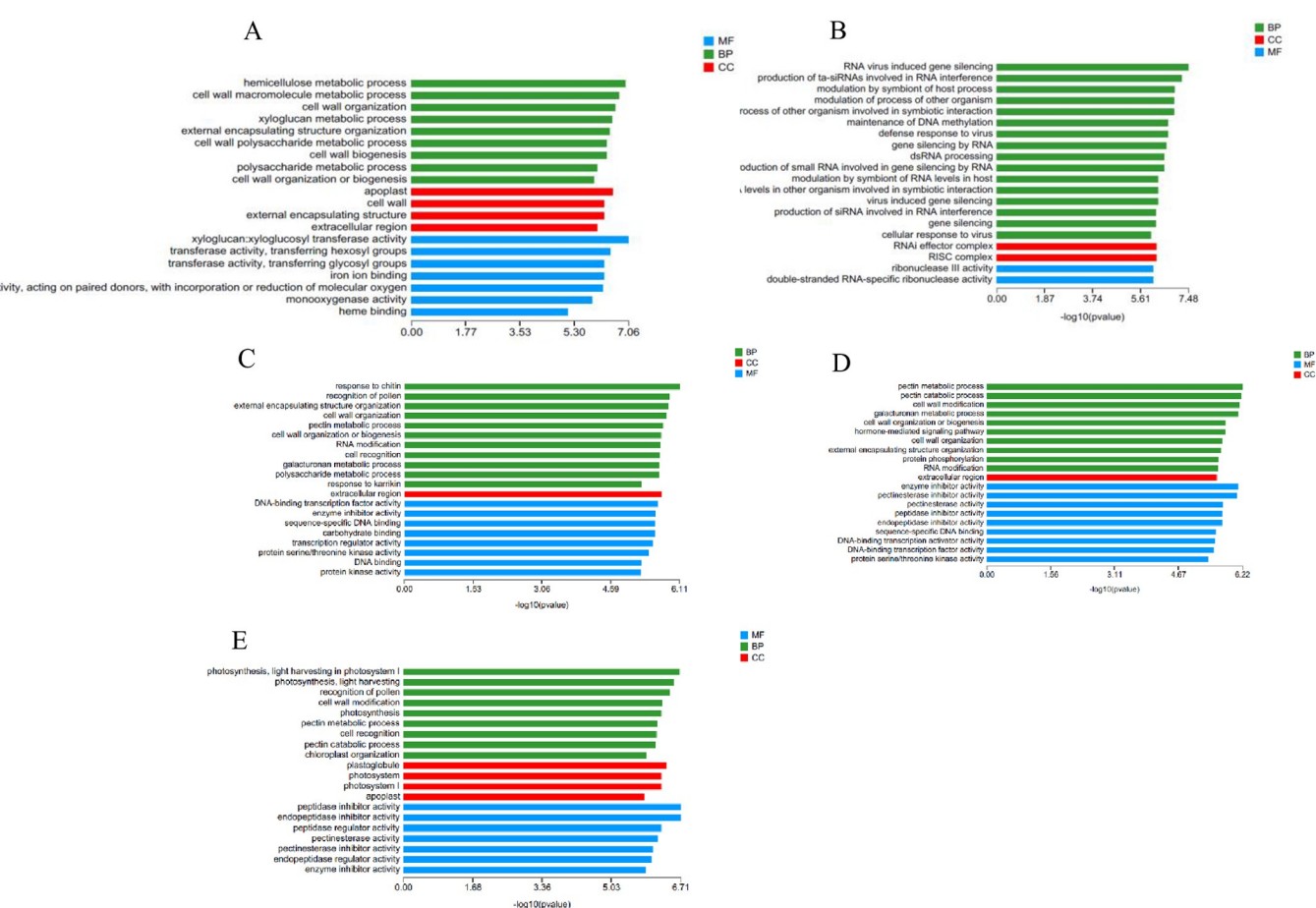

**Fig 4. GO significant enrichment analysis of DEGs.** A T2_vs_T0; B T6_vs_T0; C T12_vs_T0; D T24_vs_T0; E T48_vs_T0.

transduction, phenylpropanoid biosynthesis and carotenoid biosynthesis were the most enriched pathways. A total of 1903 differentially expressed genes in the T6_vs_T0 group were enriched in 109 pathways, of which 27 were significantly enriched. The highly enriched pathways were plant hormone signal transduction, flavonoid biosynthesis, MAPK signaling pathway—plant, phenylpropanoid biosynthesis, zeatin biosynthesis, starch and sucrose metabolism, and plant pathogen interaction. A total of 4733 differentially expressed genes in the T12_vs_T0 group were enriched in 130 pathways, among which 48 pathways were significantly enriched, such as plant hormone signal transduction, MAPK signaling pathway—plant, plant pathogen interaction, phenylpropanoid biosynthesis, starch and sucrose metabolism, and photosynthesis. The 5929 differentially expressed genes in the T24_vs_T0 group were enriched in 129 pathways, of which 26 were significantly enriched. The highly enriched pathways were plant hormone signal transduction, zeatin biosynthesis, MAPK signaling pathway—plant, diterpene biosynthesis, plant pathogen interaction, phenylpropane biosynthesis, flavonoid biosynthesis, etc. The T48_vs_T0 group had the highest number of differentially expressed genes (8659). There were 53 significantly enriched pathways, among which plant hormone signal transduction, photosynthesis—antenna protein, photosynthesis, plant pathogen interaction, MAPK signal pathway—plant, linoleic acid metabolism, and flavonoid biosynthesis were highly enriched (Fig 5). Some of the pathways in which differentially expressed genes KEGG were significantly enriched were the same in the five groups. For example, zeatin

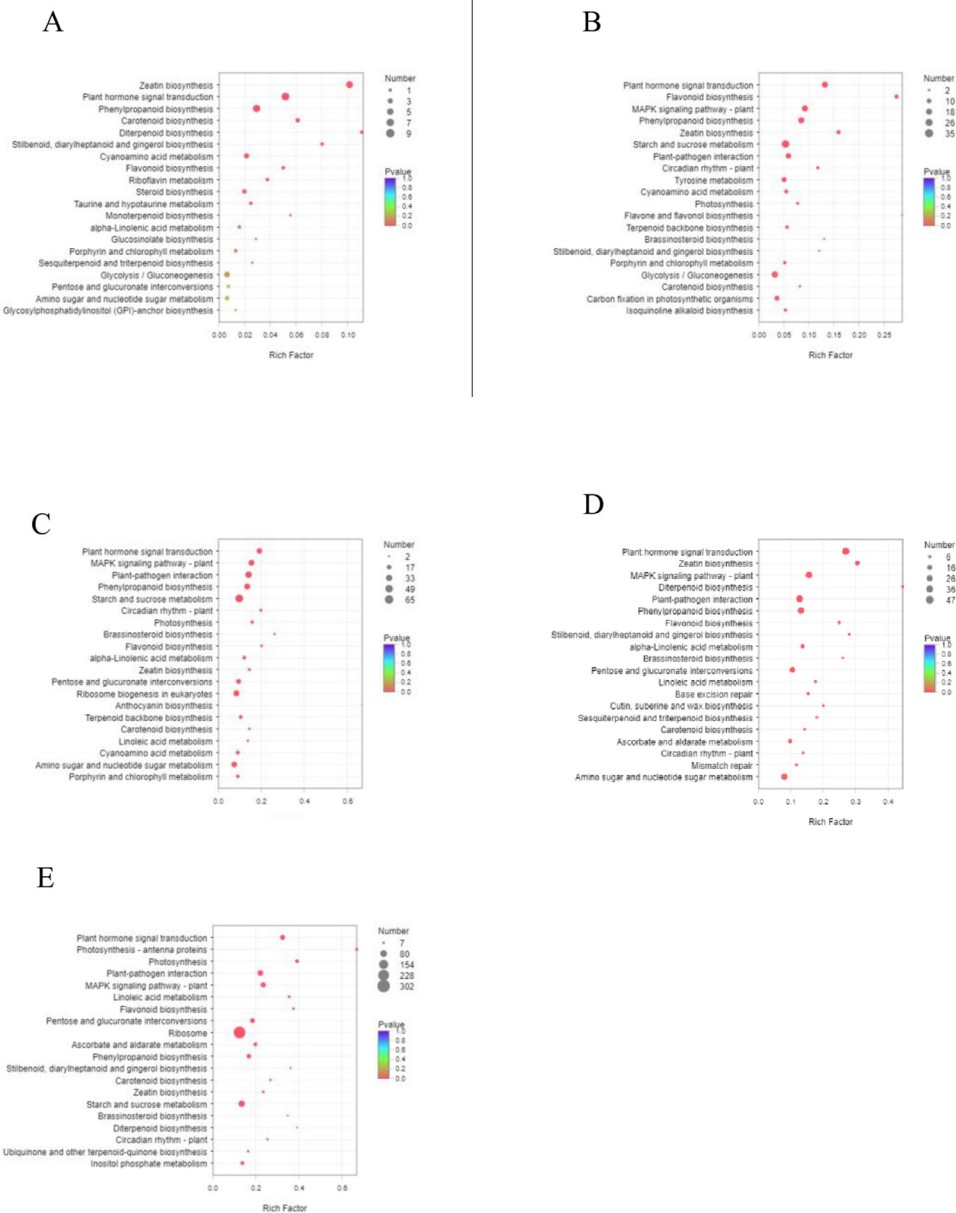

**Fig 5. KEGG enrichment analysis of DEGs.** A T2_vs_T0; B T6_vs_T0; C T12_vs_T0; D T24_vs_T0; E T48_vs_T0.

biosynthesis, plant hormone signal transduction, phenylpropanoid biosynthesis, flavonoid biosynthesis, MAPK signaling pathway—plant, starch and sucrose metabolism, and plant pathogen interaction were present in at least two groups at the same time.

## Screening of key salt-tolerance genes of *Apocynum venetum* by the WGCNA technique

Data preprocessing and construction of the sample level clustering tree. A total of 17681 genes were obtained by filtering the genes whose expression was less than 1 and coefficient of variation was less than 0.1 in all samples. Cluster analysis was conducted on all samples according

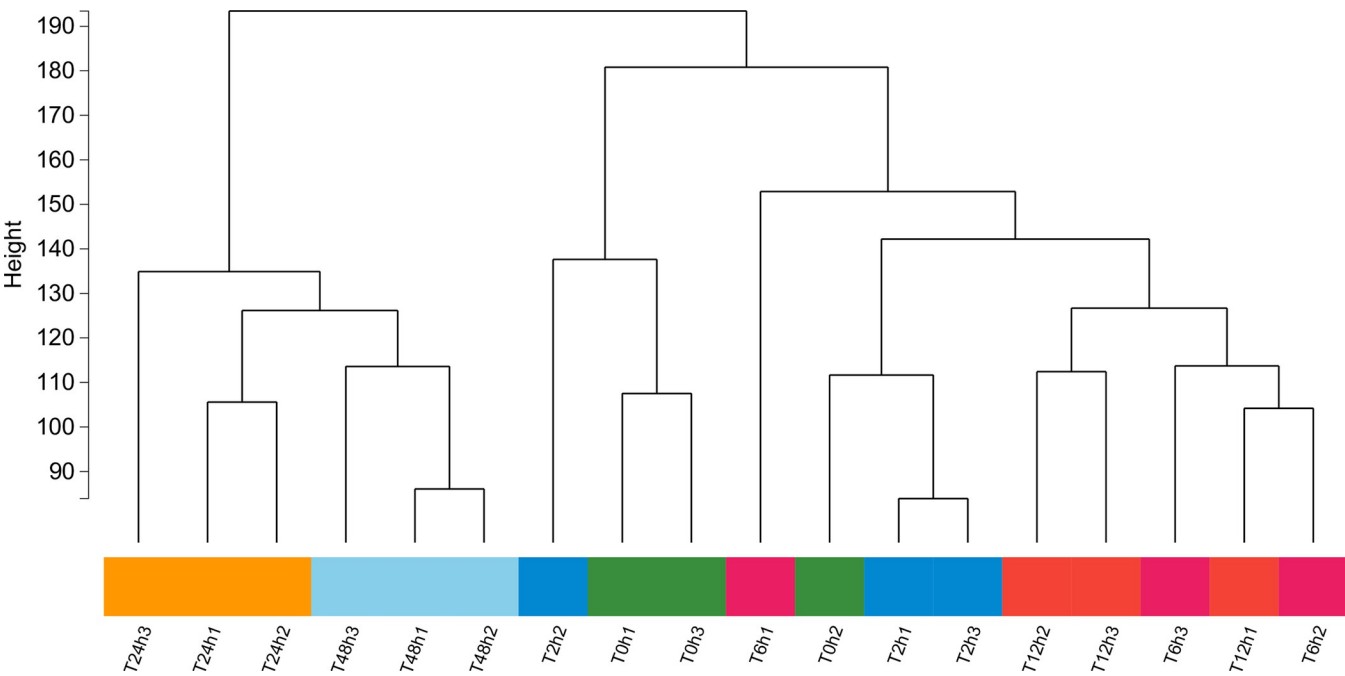

**Fig 6. Clustering dendrogram of 18 samples based on their Euclidean distance.**

to the rule of differential gene expression, as shown in Fig 6, and no outlier samples were found.

### Determining the optimal power value

One of the most critical parameters analysed by WGCNA is the power value, which mainly affects the independence and average connectivity of the co-expression module. First, the appropriate power value (β) is selected. We gradually changed the beta value to identify the optimal value, thus smoothing the average connectivity of the network. When β was 5, R2 was greater than 0.8, and the mean connectivity was higher (Fig 7). Therefore, the final beta value was determined to be 5.

### Identification of co-expressed gene modules

According to the similarity of expression patterns, the pretreated genes were divided into 12 modules (Fig 8). Each module is represented by a color, and the number of genes in each module is shown in Table 4. Among them, the turquoise module contained the most genes, with 6274 genes. The module with the smallest number of genes is the green-yellow module, with 110 genes (Table 4). The grey module refers to genes that are not divided into specific modules and will be eliminated in subsequent analysis.

### Correlation analysis between salt stress and modules

According to the similarity of expression patterns, the correlation between 12 modules and differentially expressed genes was analysed (Fig 9). Each group of data in the figure represents the correlation coefficient and significance P-value between the module and phenotype (within parentheses). The larger the absolute value, the greater the correlation, with blue indicating negative correlation and red indicating positive correlation. There were 3 modules with high

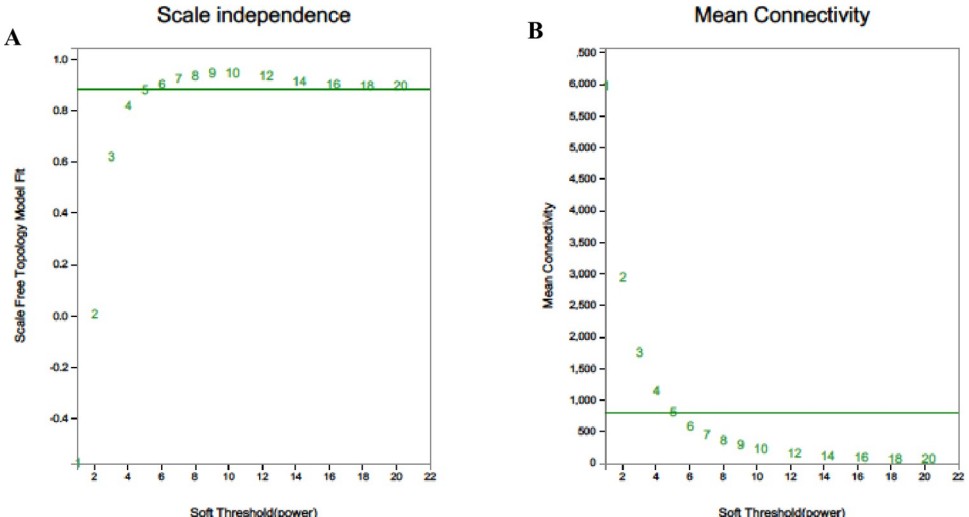

**Fig 7. Determination of the power beta value based on the adjacency matrix using WGCNA.** A for topology fitting results; B for mean connectivity.

correlation with the T2_vs_T0 group, namely, pink, rad and brown, and the correlations were 0.474, 0.618 and 0.503, respectively. The T6_vs_T0 group has two modules with high correlation coefficients, namely blue and magenta, with one positive correlation and one negative correlation, and the correlation coefficients are 0.474 and -0.445 respectively. The T12_vs_T0 group has 4 modules with high correlation, which are green-yellow, purple, turquoise and black. The first 3 modules are positively correlated, while black is negatively correlated, and the correlation coefficients are 0.474, 0.445, 0.474 and -0.445, respectively. The T24_vs_T0 group had 4 high correlation modules, 2 positive correlation modules and 2 negative

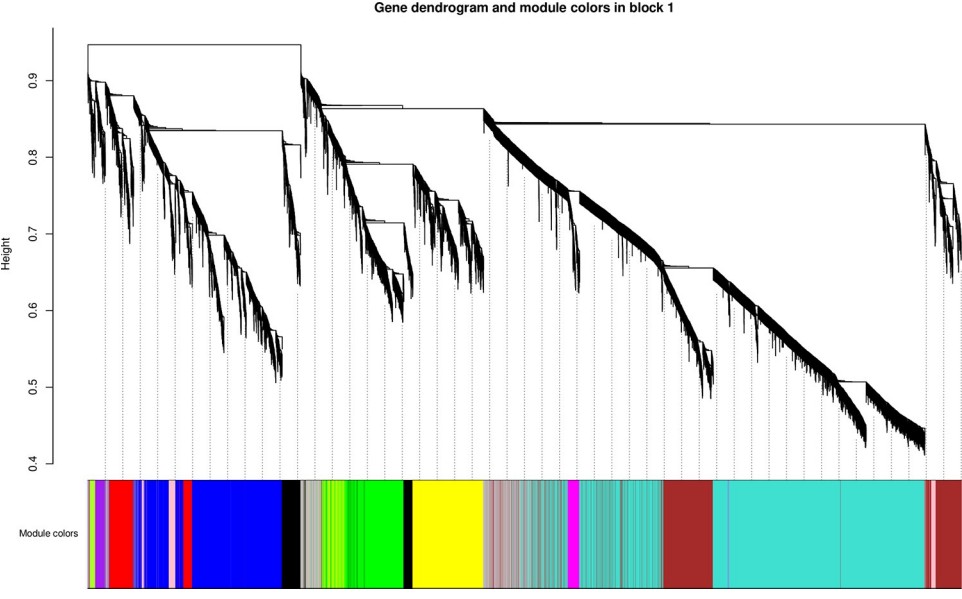

**Fig 8. Cluster dendrogram of genes in the WGCNA.** The upper part is the genes cluster dendrograms, the lower part is assigned module,the same modules had the same color.

**Table 4. Gene numbers of each module in the WGCNA.**

| Module | Number | Module | Number | Module | Number |
|---|---|---|---|---|---|
| blue | 2596 | purple | 197 | black | 640 |
| brown | 2225 | grey | 1278 | green | 1383 |
| turquoise | 6274 | yellow | 1668 | magenta | 260 |
| greenyellow | 110 | pink | 345 | red | 705 |

correlation modules. The positive correlation modules were blue and black with correlation coefficients of 0.56 and 0.618, respectively, and the negative correlation modules were brown and turquoise with correlation coefficients of -0.646 and -0.618, respectively. The T48_vs_T0 group has three modules with high correlation coefficients, namely green-yellow, brown and black, and the correlation coefficients are -0.56, 0.532 and -0.474.

We further carried out GO enrichment analysis on genes in 11 modules except the grey module and found that pathways related to salt stress, such as water transmembrane transporter activity (GO:0005372), antiporter activity (GO:0015297), flavonoid metabolism process (GO:0009812) and response to reactive oxygen species (GO:0000302), were mainly concentrated in the red, black and brown modules (Fig 10). In addition, red and black are also the two modules with the most significant correlation coefficients. Based on correlation and functional enrichment, it is speculated that there may be key genes related to salt stress in these three modules.

## Co-expression network analysis of key modules and identification of core genes

In this study, co-expression network analysis was performed for three modules of interest, red, black and brown (Fig 11). By analysing the expression trend of genes in the red module, we found that the expression level of genes in the red module was the highest at 12 h (Fig 11A). Through co-expression network analysis of this module, we found some core genes, for example, TRINITY_DN1704_c0_g1, TRINITY_DN102_c0_g1, TRINITY_DN2472_c0_g1, TRINITY_DN3073_c0_g1, TRINITY_DN3073_c0_g1, TRINITY_DN2527_ c0_g1 and TRINITY_DN10042_c0_g1 (Fig 11B).

The genes in the black module were specifically expressed at 24 h. Multiple core genes were detected in the black module, and each gene interacted with multiple genes. These genes include TRINITY_DN12990_c0_g3 and TRINITY_DN12120_c0_g1, which are involved in protein synthesis, and TRINITY_DN9926_c0_g1 and TRINITY_DN7962_c0_g1, which are unnamed proteins. These genes have high connectivity in the co-expression network and may play a key role in the process of salt stress (Fig 11D). There were three genes related to the NAD(P) H-quinone oxidoreductase subunit in the brown module: TRINITY_DN8119_c0_g1, TRINITY_DN6087_c0_g1 and TRINITY_DN34611_c0_g1. Two other genes, TRINITY_DN8706_c0_g1 and TRINITY_ DN2135_c0_g1, were associated with the ABC transporter (Fig 11F).

## Fluorescence quantitative PCR verification of salt tolerance-related differentially expressed genes

Nine differentially expressed genes related to glucose metabolism, photosynthesis and ion transport in response to salt stress were screened from the transcriptome data of *A. venetum*. under salt stress, as well as unknown genes significantly up-regulated in expression, which were as follows: TRINITY_DN3400_c0_g1, TRINITY_DN9926_c0_g1, TRINITY_DN7962_c0_g1, TRINITY_DN10560_c0_g1, TRINITY_DN6321_c0_g2,

# Correlation between module and trait

| | | T0 | T12 | T2 | T24 | T48 | T6 |
|---|---|---|---|---|---|---|---|
| MEgreenyellow | 110 | 0.158 (0.531) | 0.474 (0.0469) | -0.474 (0.0469) | 0.302 (0.223) | -0.56 (0.0157) | 0.101 (0.69) |
| MEpurple | 197 | -0.244 (0.329) | 0.445 (0.0642) | -0.0144 (0.955) | -0.244 (0.329) | -0.158 (0.531) | 0.215 (0.392) |
| MEblue | 2596 | -0.56 (0.0157) | -0.273 (0.273) | 0.129 (0.61) | 0.56 (0.0157) | -0.33 (0.181) | 0.474 (0.0469) |
| MEpink | 345 | -0.445 (0.0642) | -0.589 (0.0101) | 0.474 (0.0469) | 0.244 (0.329) | 0.101 (0.69) | 0.215 (0.392) |
| MEred | 705 | -0.589 (0.0101) | -0.158 (0.531) | 0.618 (0.00627) | -0.302 (0.223) | 0.187 (0.457) | 0.244 (0.329) |
| MEbrown | 2225 | 0.0144 (0.955) | -0.129 (0.61) | 0.503 (0.0334) | -0.646 (0.00378) | 0.532 (0.0231) | -0.273 (0.273) |
| MEturquoise | 6274 | 0.56 (0.0157) | 0.474 (0.0469) | -0.129 (0.61) | -0.618 (0.00627) | 0.129 (0.61) | -0.417 (0.0851) |
| MEblack | 640 | 0.158 (0.531) | -0.445 (0.0642) | -0.129 (0.61) | 0.618 (0.00627) | -0.474 (0.0469) | 0.273 (0.273) |
| MEmagenta | 260 | 0.589 (0.0101) | -0.0144 (0.955) | -0.215 (0.392) | -0.244 (0.329) | 0.33 (0.181) | -0.445 (0.0642) |
| MEgreen | 1383 | 0.302 (0.223) | -0.158 (0.531) | -0.215 (0.392) | 0.101 (0.69) | -0.244 (0.329) | 0.215 (0.392) |
| MEyellow | 1668 | 0.445 (0.0642) | 0.417 (0.0851) | -0.129 (0.61) | -0.33 (0.181) | -0.215 (0.392) | -0.187 (0.457) |
| MEgrey | 1278 | -0.187 (0.457) | 0.417 (0.0851) | 0.273 (0.273) | -0.244 (0.329) | -0.33 (0.181) | 0.0718 (0.777) |

**Fig 9. Correlation between modules and different durations of salt stress.** In the heatmap of correlations, negative number (blue) indicates negative correlation between eigengenes and modules, and positive number (red) indicates positive correlation.

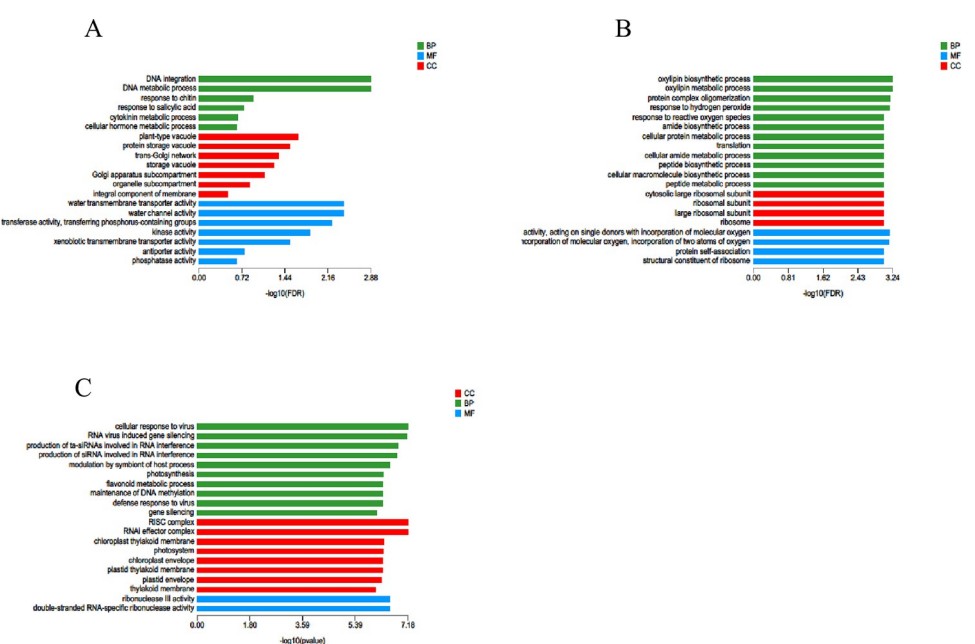

**Fig 10. GO functional classification of genes in key modules.** A red module; B black module; C brown module.

TRINITY_DN244_ c0_g1, TRINITY_DN10153_c0_g1, TRINITY_DN26308_c0_g1, TRINITY_DN10042_c0_g1.

To verify the accuracy of the transcriptome data of *A. venetum*, fluorescence quantitative verification analysis was performed for these 9 genes (Fig 12). The results showed that the expression trend of 9 differentially expressed genes at different stress times was consistent with that of transcriptome sequencing. Therefore, fluorescence quantitative analysis showed that the transcriptome sequencing results were accurate, and the 9 genes screened were preliminarily identified as the key genes in response to salt stress in *A. venetum*.

## Discussion

The plant response to salt stress is a complex regulatory process. At the molecular level, plants respond through significant rearrangement of the transcriptome and the induction of many stress-responsive genes. Stress-responsive genes are involved in encoding the production of protective metabolites, transporters, and antioxidant enzymes that help plants survive and grow during stress [16]. In this study, high-throughput sequencing technology was used to sequence the transcriptome of *A. venetum* under salt stress at 2 h, 6 h, 12 h, 24 h, and 48 h, and a number of differentially expressed genes (DEGs) involved in the response to salt stress, including oxidoreductase activity, polysaccharide metabolism, phytohormone signalling, phenylpropanoid biosynthesis, flavonoid biosynthesis, and photosynthesis, were identified (Figs 4 and 5). Sugar metabolites can stabilize cell membranes and protoplasts, enhance osmotic potential to resist salt stress, and play a role in supplying energy and carbon sources when photosynthesis is limited [17]. Glucosyltransferases play a variety of roles in cellular metabolism. In *A. thaliana*, which continuously expresses the UDP-glucosyltransferase family member *UGT85U1*, salt stress tolerance is enhanced by altering the expression of genes related to hormone signalling and stress response under salt stress [18]. Studies have shown that overexpression of *GhJAZ2* in cotton can inhibit jasmonic acid (JA) signalling and reduce cotton tolerance to salt stress [19]. The wheat *TaPP2C1* gene negatively regulates abscisic acid (ABA) signalling

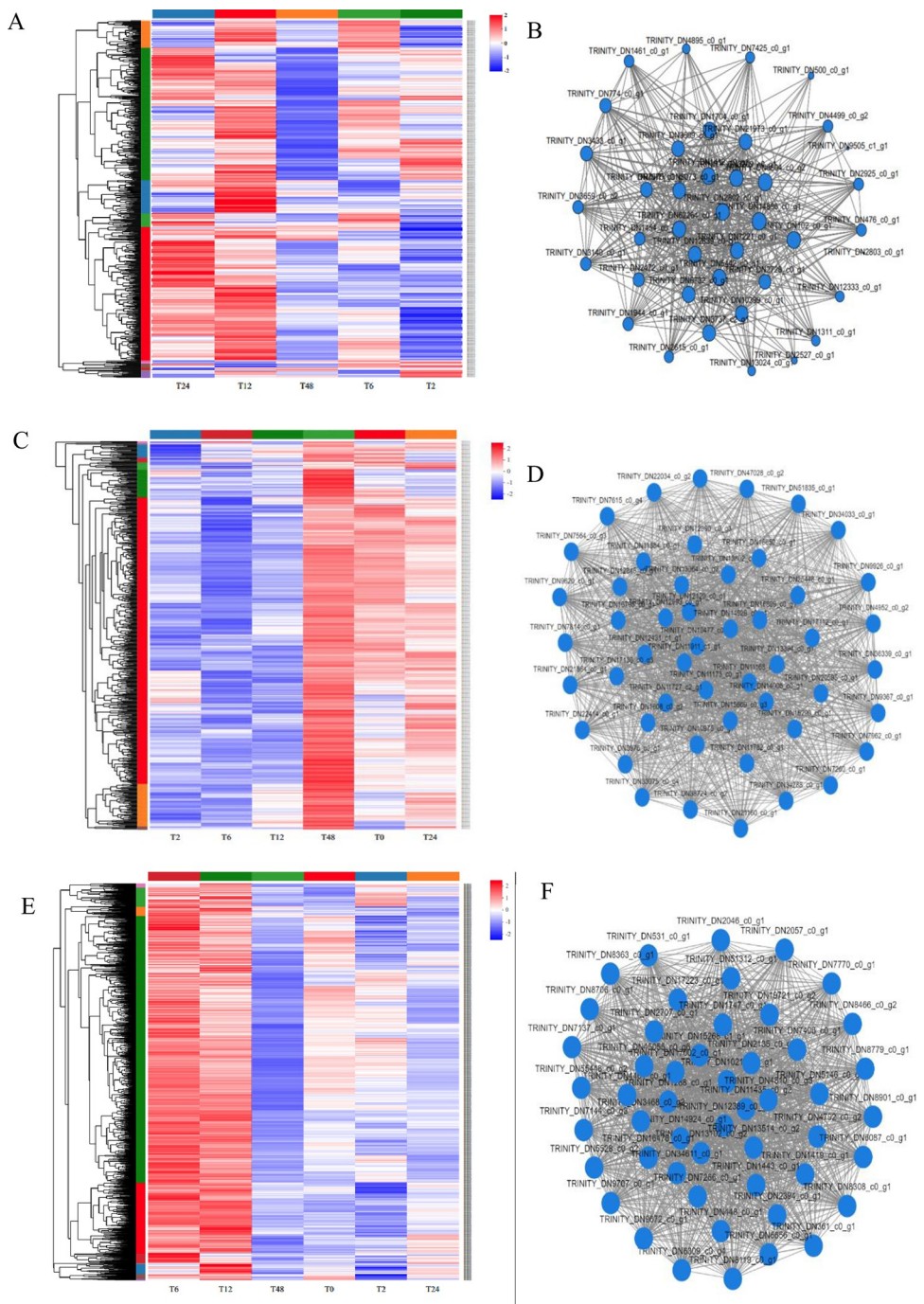

**Fig 11. Gene expression profile and co-expression network analysis of key modules. A**, **C**, and **E** heatmaps show the expression patterns of genes in the red, black, and brown modules, respectively. **B**, **D**, **F** represent the co-expression network analysis of genes in the red, black and brown modules. The size of node circle is positively correlated with the number of the interacting genes. In the visualization graph, each node represents a gene, and usually the greater the node connectivity (i.e. the greater the number of connected nodes, i.e. the more radiating edges), the more important it is.

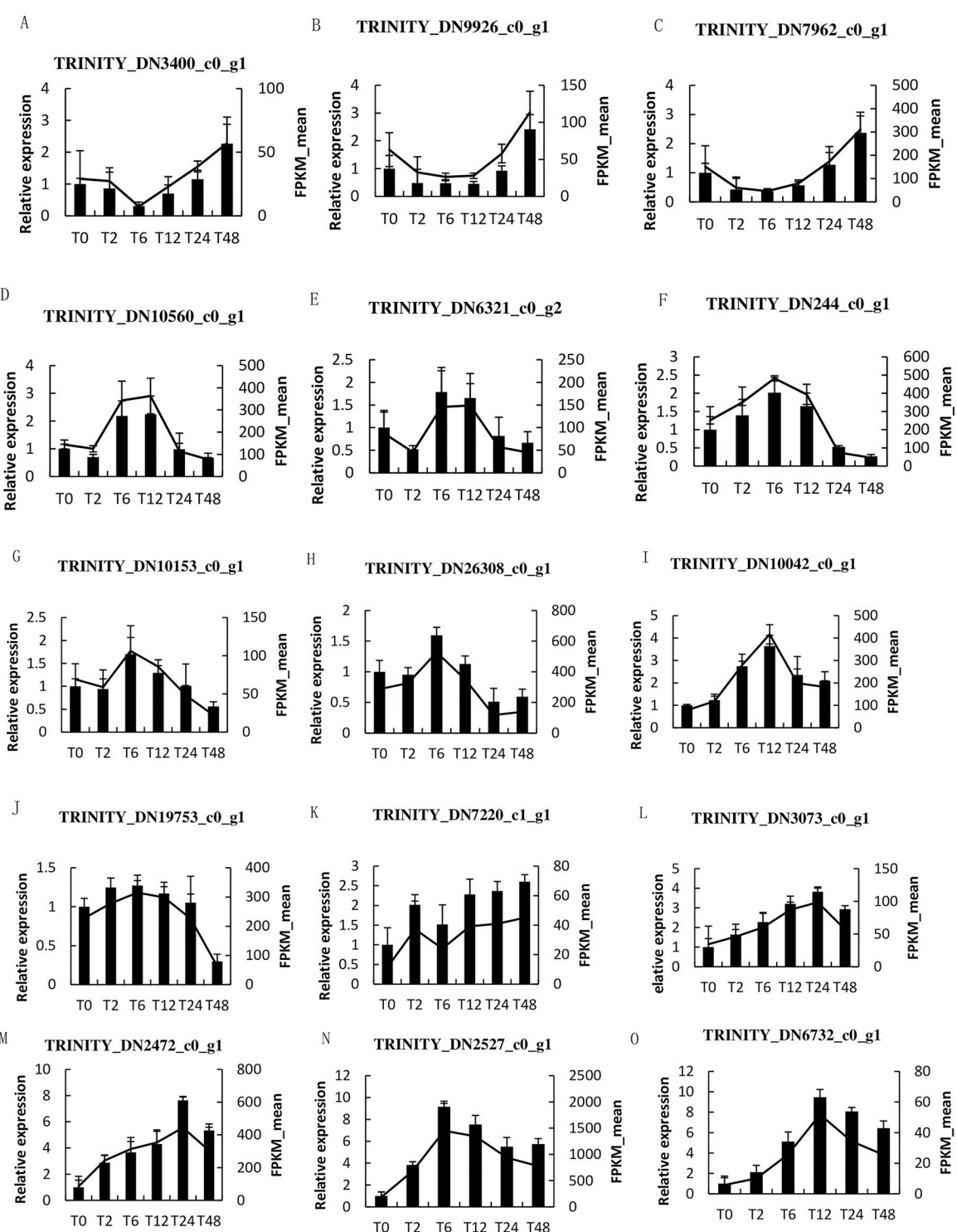

**Fig 12. qRT-PCR analysis of 9 salt tolerance-related differentially expressed genes.** A TRINITY_DN3400_c0_g1,sucrose synthase; B TRINITY_DN9926_c0_g1,unnamed protein; C TRINITY_DN7962_c0_g1,hypothetical protein CDL12_15788; D TRINITY_DN10560_c0_g1,photosystem II core complex proteins; E TRINITY_DN6321_c0_g2,aquaporin PIP2-7; F TRINITY_DN244_c0_g1,aquaporinPIP2-5; G TRINITY_DN10153_c0_g1,uncharacterized protein; H TRINITY_DN26308_c0_g1, ATPsynthase; I TRINITY_DN10042_c0_g1,ABC transporter. The bar in the figure represents the relative expression level, and the line represents FPKM mean.

but positively regulates salt tolerance [20]. Flavonoids are closely related to the salt stress tolerance of plants. Under salt stress, the flavonoid biosynthesis and phenylpropanoid biosynthesis pathways in *A. venetum* were significantly enriched. Salt stress induces the expression of flavonoid biosynthesis genes and proteins, which can effectively enhance the salt tolerance of *A. venetum* [13].

In this study, the WGCNA method was used to analyse the DEGs in *A. venetum* under salt stress, and three modules closely related to salt stress were found, namely, the red, black, and brown modules. Furthermore, core genes that are closely related to salt tolerance were predicted by protein-protein interaction network (PPIN) analysis. The heat stress transcription factor (HSF) in the red module was identified as the core gene. HSFs play a crucial role in plant responses to various abiotic stresses by regulating the expression of stress-responsive genes such as heat shock proteins (*Hsps*) [21]. *FvHsfA2a* in the Hsf gene identified in WT diploid woodland strawberry was significantly upregulated under salt stress treatment to improve the salt tolerance of plants [22]. The expression of 19 *PsnHSF* genes was upregulated in poplar after 24 h of salt stress, indicating that *PsnHSF* genes play an important role in the response to high salt stress [23]. In this study, the TRINITY_DN6732_c0_g1 gene was identified as an HSF, and its expression increased significantly after salt stress treatment, peaked at 12 h, and then gradually decreased, but its expression was still significantly higher than that of the control at 48 h. Therefore, it is speculated that this gene plays an important role in coping with salt stress.

The serine carboxypeptidase-like (SCPL) gene family plays an important role in the defense response of plants to abiotic stress and can regulate the defense mechanism of plants against adversity to resist the damage caused by adverse stress [24]. This study identified the TRINITY_DN102_c0_g1 gene in the red module, which encodes serine carboxypeptidase, and its expression increased significantly with stress treatment time, which is consistent with the increase in the expression of the serine carboxypeptidase gene (*ZmSCP*) in maize induced by salt stress [25]. The drought tolerance and high temperature tolerance of rice plants overexpressing the *OsSCP* gene were significantly higher than those of WT plants, indicating that the *OsSCP* gene plays an important role in improving rice drought tolerance and high-temperature tolerance [26]. In addition, this study also identified the TRINITY_DN3073_c0_g1 gene, which encodes a serine/threonine protein kinase, in the red module. Expression of this gene was significantly increased after salt stress and peaked at 48 h of stress, indicating that salt stress induces its expression to maintain cell ion homeostasis, which is possibly related to the specific involvement of the serine/threonine protein kinase signalling pathway in the response to ion stress [27].

The core genes in the black module are mainly involved in protein synthesis, including the unknown proteins TRINITY_DN9926_c0_g1 and TRINITY_DN7962_c0_g1, which are possible new genes in *A. venetum*.

When plants are subjected to abiotic stress, carbon assimilation is inhibited, and the energy absorbed by plants cannot be fully utilized. Excessive energy accumulation causes the photosystem I (PSI) and photosystem II (PSII) reaction centres in chloroplast thylakoids to accumulate ROS, which can cause photoinhibition and photodamage in plants [28]. Under stress conditions, PSII is very sensitive to ROS and is easily damaged by ROS [29]. In this study, a core gene, TRINITY_DN10560_c0_g1, was identified in the brown module and annotated as a PSII complex protein, and its expression fluctuated significantly, i.e., the expression decreased after 2 h of stress, increased at 6 h, decreased at 12 h, and reached a minimum at 48 h. Therefore, it is inferred that this gene is suppressed during salt stress. The genes involved in encoding aquaporins (TRINITY_DN6321_c0_g2 and TRINITY_DN244_c0_g1) identified in the brown module were upregulated. Aquaporins are involved in osmoregulation in plant cells,

improve the hydraulic conductivity of roots and leaves, enhance transpiration, and effectively improve plant water use efficiency under abiotic stress conditions [30–32]. Transgenic bananas overexpressing *MusaPIP2;6* exhibited better photosynthetic efficiency and lower membrane damage under salt stress [33]. Overexpression of the ginseng (*Panax ginseng*) aquaporin *PgTIP1* in *A. thaliana* showed enhanced tolerance to salt and drought stresses [34]. Therefore, it is inferred that these genes may also have a similar regulatory mechanism in *A. venetum*.

## Conclusion

A total of 11914 differentially expressed genes were identified by comprehensive analysis of the genes and transcriptome expression profiles in response to salt stress in *A. venetum*. Among them, 373 differentially expressed genes were identified in group T2_vs_T0, 1903 differentially expressed genes were identified in group T6_vs_T0, 4733 differentially expressed genes were identified in group T12_vs_T0, 5929 differentially expressed genes were identified in group T24_vs_T0, and 8659 differentially expressed genes were identified in group T48_vs_T0. GO and KEGG databases were used for enrichment analysis. The significant enrichment pathways were as follows: REDOX enzyme activity, polysaccharide metabolism, plant hormone signal transduction, phenylpropane biosynthesis, flavonoid biosynthesis and photosynthesis.

In this study, the WGCNA method was used to construct the gene hierarchical clustering tree, divide the modules, and finally obtain 12 modules according to the similarity of module expression patterns. Red, black and brown were identified as the core modules that regulate the tolerance to salt stress according to the correlation analysis between modules and differentially expressed genes and the functional enrichment analysis of GO genes in modules. In the core module of screening, according to the correlation between genes, the co-expression regulatory network is constructed to identify the core genes in the module. Finally, the genes TRINITY_DN102_c0_g1, TRINITY_DN3073_c0_g1 and TRINITY_DN6732_c0_g1 related to serine carboxypeptidase, SOS signaling pathway and heat shock transcription factor in the red module were determined to be the core genes in the red module. The core genes of the black module are unknown proteins TRINITY_DN9926_c0_g1 and TRINITY_DN7962_c0_g1, which may be new genes in *A. venetum*, and their roles in response to salt stress need to be further explored.The genes in the brown module were mainly concentrated in photosynthesis and osmotic balance. TRINITY_DN6321_c0_g2 and TRINITY_DN244_c0_g1 encode aquaporins, which help to maintain the water balance of cells and protect the survival of Rhizomonas arum under stress conditions.

## Materials and methods

### Instruments

Centrifuge (Eppendorf, Centrifuge 5418, Germany); UV Spectrophotometer (Thermo,NanoDrop2000, USA); Quantometer (Invitrogen,Qubit2.0, USA); Magnetic Grate(Invitrogen, Magnetic stand-9,USA); Bioanalyzer (Aglient,2100, USA).

### Kits

TruSeq Stranded mRNA LT Sample Prep Kit (Illumina,USA,Cat.No.: RS-122-2101); AgencourtAMPure XP (BECKMAN COULTER, USA,Cat.No.:A63881); QubitRNA Assay Kit (LifeTechnologies, USA,Cat.No.:Q32852); Bioanalyzer 2100 RNA-6000 Nano Kit (Aglient, USA,

Cat.No.:5067–1511); Bioanalyzer 2100 DNA-1000 Kit (Aglient, USA, Cat.No.:5067–1504); SuperScript II Reverse Transcriptase (Invitroge,USA,Cat.No.: 18064014)

## Test materials

Annual *A. venetum* seedlings with consistent growth were selected from the Ordos A. venetum plot (108˚44'44" east longitude, 40˚35'23" north latitude) in the Inner Mongolia Autonomous Region. The test samples were included in the national north forage germplasm resource medium term library (No. 16797), and the national uniform number of this species was CF057769. The seedlings were transported back to the laboratory in 1/2 Hoagland's nutrient solution. After 3 days of acclimatization in the laboratory, plants with good growth were selected to the salt stress treatment. According to the pre-experiment, 300 mmol/L was selected as the stress concentration for the transcriptome test. The roots of the test group were carefully cleaned in pure water, and then transferred to the stress nutrient solution (containing 300 mmol/L NaCl). After 0 h, 2 h, 6 h, 12 h, 24 h and 48 h of salt stress, *A. venetum* leaves were quickly paaced into frozen tubes for liquid nitrogen deep freezing, and then the samples were stored in the refrigerator at -80˚C for reserve. To ensure the ability to conduct statistical analyses, three biological repli-cates were used for the transcriptome sequencing analysis and three biological replicates were used for qRT-PCR analysis.

## RNA extraction

Total RNA was extracted from leaf samples of *A.venetum* after salt stress using mirVana™ miRNA ISOlation kits. Agilent 2100 (Agilent Technologies, USA) was used to determine the integrity of total RNA quality, and NanoDrop 2000 (Nanodrop Thermo Scientific, USA) was used to determine the purity and concentration of RNA. High quality RNA was used to construct sequencing libraries (A260/280$\geq$2.0, A260/230$\geq$1.0, RIN$\geq$ 7,28 S/18S$\geq$1.0).

## Construction and sequencing of the cDNA library

After digesting DNA with DNase, Poly(A)mRNA was enriched with magnetic beads with Oligo(dT). The mRNA was broken into short fragments by adding reagents, and the interrupted mRNA was used as a template to synthesize the first strand cDNA with six-base random primers. The second strand cDNA was synthesized by adding dNTPs, ribonuclease H, DNA polymerase I, etc., and the double-strand cDNA was purified by kits.The purified double-stranded cDNA was end-repaired, poly(A) was added, and the sequencing joint was connected. The cDNA library was constructed by PCR amplification and sequenced by an illumina novaseq 6000 sequencer.

## Transcriptome de novo assembly and functional notes

To obtain high-quality reads that could be used for subsequent analysis, quality filtering of raw reads generated from high-throughput sequencing was needed. First, Trimmomatic software was used for quality control, and the joint was removed. On this basis, low quality bases (mass < 20) and N bases (N content > 5%) were filtered out to obtain high-quality clean reads. Using the concatenation method of paired-end in Trinity software (version: 2.4), clean reads were concatenated to obtain a Transcript sequence. According to sequence similarity and length, the longest sequence was selected as Unigene, which was used as reference sequence for subsequent analysis. Diamond software was used to compare Unigene to the NR, KOG, GO, Swiss-Prot, eggNOG and KEGG databases, and HMMER software was used to compare Pfam database for functional analysis of Unigene. This article uses twice GO

enrichment analyses, one for differentially expressed genes at different stress times, and the other for analyzing genes in key modules obtained from WGCNA analysis.

## Identification of differentially expressed genes

FPKM and read counts value of each unigene was calculated using bowtie2 (Version 2.3.3.1) and eXpress (Version 1.5.1). Gene expression was calculated using (FPKM)(Fragments Per Kilo bases per Million fragments) method. DESeq software (Version 1.18.0) was used to standardize the number of counts of each sample gene (BaseMean value was used to estimate the expression), the difference multiples were calculated, and NB (negative binomial distribution test) was used to test the significance of the difference between read numbers. Hierarchical cluster analysis of DEGs was performed to explore transcripts expression pattern. GO enrichment and KEGG pathway enrichment analysis of DEGs were respectively performed using R based on the hypergeometric distribution. Finally, differential genes were screened out according to the difference multiples and significance test results. The screening conditions for differentially expressed genes were that the difference factor was greater than 2 and the FDR value was less than 0.05.

## Fluorescence quantitative PCR verification of differentially expressed genes

Nine differentially expressed genes related to salt tolerance were screened from the transcriptome sequencing data of *A. venetum* for fluorescence quantitative PCR verification (Table 5). The PerfectStartTM Green qPCR SuperMix kit was used on a LightCycler® 480II fluorescence quantitative PCR instrument (Roche, Swiss) with three biological replicates per sample. Fluorescence quantitative reaction system: 2×PerfectStartTM Green qPCR SuperMix, 5 µl; 10 µM Forward primer, 0.2 µl; 10 µM Reverse primer, 0.2 µl; cDNA, 1µl; Nuclease-free $H_2O$, 3.6 µl. The PCR procedure was as follows: 94˚C for 30 s; 45 cycles of 94˚C for 5 s and 60˚C for 30 s. The effectiveness of primers was determined according to the Ct value of qPCR amplification and the dissolution curve of primers. Actin was selected as the endogenous reference gene, and the relative expression level of the target gene was calculated by the 2-ΔΔCt method.

## Weighted gene co-expression network analysis (WGCNA)

WGCNA is a systematic biological method. It can be used to explore the relationship between genes and related traits by the identification of co-expressed modules and hub genes in the network. After data preprocessing, genes are classified and those with similar expression patterns

**Table 5. Primer information.**

| Gene symbol | Forward primer (5' to 3') | Reverse primer (5' to 3') | Product length (bp) |
|---|---|---|---|
| TRINITY_DN3400_c0_g1 | CTCAATAGGCACCTTTCAGC | TCATTGTCTTGCCCTTGTAGT | 100 |
| TRINITY_DN9926_c0_g1 | GTCGTTTCTTGATCATCTCCAC | TGTAGTCTCATCATCAACCTCG | 97 |
| TRINITY_DN7962_c0_g1 | CAAGAAGTGTCAGTGTAGGC | CCTATTCAGAGTTTCTTCACCA | 80 |
| TRINITY_DN10560_c0_g1 | ATTGCTGAAGGCAGCGATAA | TAAGGGCTGGTTGGAAGAT | 97 |
| TRINITY_DN6321_c0_g2 | GACCAGTGTGACGGAGTT | AGGGTTGATATGTCCACCT | 108 |
| TRINITY_DN244_c0_g1 | GTTACAGCACCGGAGTTG | TCTTGGGATCAGTGGCAGA | 87 |
| TRINITY_DN10153_c0_g1 | TTGGTGATGCTGACCAAGT | CCCTTCTGGATTTGTATCCGTA | 92 |
| TRINITY_DN26308_c0_g1 | CTCTTTCTGACTCCTCGG | TTCGTGGTATTGGTCGAGTTAT | 85 |
| TRINITY_DN10042_c0_g1 | AGTTGATTACAAGCGGGC | AAACAGTGACACCCTTATAGC | 104 |
| *Apocynum venetum* actin | TTATCATTACCTACACCACCTCC | TGTAAGACTGAAGCCATCTCG | 135 |

are classified into one class, which is called a module. The Identification of co-expressed gene modules is to classify genes with similar expression patterns into a group, and such a group of genes is called modules.The gene co-expression network is a scale-free weighted gene network. Unweighted network means that the correlation between genes can only be 0 or 1, where 0 indicates that the two genes are not connected, while 1 indicates that there is a connection between genes. Weighted network refers to the fact that genes are not only related or not, but also record their correlation values, which are the weights (correlations) of the relationships between genes. In the co-expression network, the expression of each gene at a specific time or space is considered as a node. In order to obtain the correlation between genes, it is necessary to calculate the person coefficient between any two genes. To determine whether the expression profiles of the two genes are similar, WGCNA adopts a method based on soft threshold(β power), which means that the correlation coefficient between genes is obtained through weighting rather than artificially setting, can effectively avoid the limitations of artificially setting.The parameter settings for module recognition are: soft threshold (β power) is 5, the minimum number of genes that make up the module is 30, and the minimum correlation between module members and vector genes is 0.3.Remove modules below a certain distance threshold, that is, merge modules with smaller distances, and the threshold is 0.25.Using spearman rank correlation coefficient to analyze correlation analysis between salt stress and modules.The DEGs were further divided into twelve modules and correlation of each module with salt stress was calculated.Obtain the hub genes of the module through visual network analysis. Analyze the connections with the top 50 connectivity nodes within the module and weights greater than 0.3 between nodes to obtain the core genes in the network. Each node represents a gene, which is connected to a different number of genes. The gene which is connected to a greater number of genes is denoted with a bigger size and is more important for its interaction with a large number of genes.

## Acknowledgments

We thank OE biotech Co., Ltd. (Shanghai, China) for providing technical support.

## Author Contributions

**Conceptualization:** Xi Zhen, Tao Wan.

**Data curation:** Xuyang Liu, Xiaoming Zhang, Shujie Luo, Wencheng Wang.

**Formal analysis:** Xi Zhen, Shujie Luo, Wencheng Wang.

**Investigation:** Xi Zhen, Xuyang Liu, Tao Wan.

**Methodology:** Xiaoming Zhang, Tao Wan.

**Supervision:** Xiaoming Zhang.

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
