## [Decision Letter · Decision Letter 0]

30 Oct 2023

PONE-D-23-25332Identification of core genes involved in the response of Apocynum venetum to salt stress based on transcriptome sequencing and WGCNAPLOS ONE

Dear Dr. Zhen,

Thank you for submitting your manuscript to PLOS ONE. After careful consideration, we feel that it has merit but does not fully meet PLOS ONE’s publication criteria as it currently stands. Therefore, we invite you to submit a revised version of the manuscript that addresses the points raised during the review process.

We look forward to receiving your revised manuscript.

Kind regards,

Mohammad Sadegh Taghizadeh

Academic Editor

PLOS ONE

Reviewers' comments:

Reviewer's Responses to Questions

**Comments to the Author**

1. Is the manuscript technically sound, and do the data support the conclusions?

Reviewer #1: Yes

Reviewer #2: Yes

Reviewer #3: Yes

Reviewer #4: Yes

2. Has the statistical analysis been performed appropriately and rigorously? 

Reviewer #1: Yes

Reviewer #2: Yes

Reviewer #3: No

Reviewer #4: Yes

3. Have the authors made all data underlying the findings in their manuscript fully available?

Reviewer #1: Yes

Reviewer #2: Yes

Reviewer #3: Yes

Reviewer #4: Yes

4. Is the manuscript presented in an intelligible fashion and written in standard English?

Reviewer #1: Yes

Reviewer #2: Yes

Reviewer #3: Yes

Reviewer #4: Yes

5. Review Comments to the Author

Reviewer #1: Comments to the Author

1.Is the manuscript technically sound, and do the data support the conclusions?

Yes.

2.Has the statistical analysis been performed appropriately and rigorously?

Yes. In correlation analysis, it is better to provide a significant difference analysis in addition to the correlation coefficient

3.Have the authors made all data underlying the findings in their manuscript fully available?

Yes.

4.Is the manuscript presented in an intelligible fashion and written in standard English?

The English writing is good, but the article needs further careful proofreading to avoid typos.

Review:

Abstract

1. Methods of salt stress are briefly emphasized in the abstract

Some conjunctions can be omitted if the number of words is limited, like “Herein ，Finally”. Please check the presentation carefully.

This sentence: “TRINITY_DN102_c0_g1, TRINITY_DN3073_c0_g1 and TRINITY_DN6732_c0_g1 related to serine carboxypeptidase, SOS signaling pathway and heat shock transcription factor in the red module were determined to be the core genes in the red module” , the phrase of “in the red module” appear two times.

Introduction

1.Please check the statement of the article, some sentences are repeated. For example“In summary, scholars in China and around the world have carried out many studies on the medicinal components, genetic diversity, and salt tolerance of A. venetum, but the molecular mechanism of the salt tolerance of A. venetum is still unclear, and more in-depth research is needed. ”

Because of the previous similar description, here can be rephrased.

Like: Although there are some research results inthe medicinal components, genetic diversity, and salt tolerance of A. venetum, it is very necessary to conduct in-depth research in the molecular mechanism of the salt tolerance of A. venetum.

Results

1.There are some mistakes of words. Like: “All indexes of the original transcriptome sequencing data met the data requirements and couble be spliced and assembled in the next step”.

2.In Table 1, there are two columns of “raw reads”

3.In the part of Correlation analysis, does the value of the correlation coefficient represent the strength of the correlation? Please state them in the notes of Fig or in the corresponding paragraph.

4.Please state them in the notes of Figures, like Fig8, 9, 11,12, What do dots, lines or columns represent in the picture?

5.In the results section of the qRT-PCR analysis, Figure 12 displayed qRT-PCR results for 15 genes, but only results for 9 genes are stated in the text. This discrepancy should be rectified to ensure consistency between the number of genes presented in the figure and those discussed in the text.

Discussion

1.In the concluding section, the author mentions "TRINITY_DN9926_c0_g1 and TRINITY_DN7962_c0_g1, which may be new genes in A. pseudotensis, and their roles in response to salt stress need to be further explored." It appears that these new genes were discussed in the context of A. pseudotensis, while the experimental material used was Apocynum venetum.

2.“The higher the connectivity of one gene in the WGCNA is, the higher the degree of connection between this gene and other genes and the higher the importance of this gene; such a gene is usually a core gene. ” whose opinion is it? Please add the references or explain it.

Materials and methods

1.Is this a repetition?

Wild A. venetum plants with good growth were divided into two groups, the salt stress

treatment group and the control group, and each treatment was repeated 3 times. According to the pre-experiment, 300 mmol/L was selected as the stress concentration for the transcriptome test.

Wild Apocaf plants with good growth were divided into two groups, the salt stress treatment group and the control group, and each treatment was repeated 3

times.

2.Please add the collection site or planting site. If it is wild, how to select the salt stress test site and control test site? If it is grown, what salt you added and how much, please describe clearly and briefly in the part of Abstract.

3.How many test treatments? How many repetitions per process? What is the sample size used in the test?

4.In the transcriptomics section, it is essential to specify whether the sequencing work was conducted by a sequencing company. If so, this information should be explicitly stated in the manuscript. Providing this detail in the article is crucial for transparency and reproducibility, as it allows other researchers to understand the data generation process and the involvement of an external sequencing service provider.

5.The software or analysis platform， like "DESeq software" requires a version number or a reference.

6.Why did you choose these methods in your research, like WGCNA technique. So please state them in the part of Methods.

7.This part of “Screening of key salt-tolerance genes of Apocynum venetum by the WGCNA technique… ”, is it misplaced? The result is not described here

8.It is shown in the result that Go analysis was done twice, why did you do so, please state in the method.

9.How to do the “Co-expression network analysis”? please state in the method.

Reviewer #2: The manuscript entitled “Identification of core genes involved in the response of Apocynum venetum to salt stress based on transcriptome sequencing and WGCNA” has many mistakes, and the authors need to rectify many portions.

• "The adaptation of plants to salinization is the joint result of a series of complex regulatory mechanisms, e.g., morphological, physiological, and genetic traits [1]." - Could you rephrase the sentence to clarify the role of these complex regulatory mechanisms in plant adaptation to salinity?

• "Under high-salt stress (high concentrations of salt), plant growth and development may be inhibited, and death is even possible." - What specific salt concentrations constitute "high-salt stress," and how does it affect plant growth and development?

• "The presence of a large amount of Na+ and Cl- could cause ion toxicity and nutrient deficiency." - Can you explain how the presence of Na+ and Cl- ions leads to ion toxicity and nutrient deficiency in plants?

• "Most studies on the adaptation mechanisms of A. venetum under salt stress focus on physiological and phenotypic aspects, and some studies try to clarify the salt tolerance mechanism of A. venetum at the molecular level." - What have these studies revealed about the adaptation mechanisms of A. venetum to salt stress, both at the physiological and molecular levels?

• "Due to the accumulation of flavonoids, transgenic Arabidopsis thaliana showed higher salt tolerance than wild-type (WT) A. thaliana [11]." - Can you explain the relationship between flavonoid accumulation and salt tolerance in transgenic Arabidopsis thaliana?

• "The A. venetum DEAD-box helicase gene (AvDH1), stably inherited in the cotton genome, significantly improved the salt tolerance of transgenic cotton lines [12]." - What is the role of the AvDH1 gene in enhancing salt tolerance, and how does it interact with the cotton genome?

• "This study analysed the transcriptome dataset of A. venetum under salt stress at six time points and identified core salt stress-responsive genes." - Can you provide more details about the methodology used in this analysis and the significance of identifying core salt stress-responsive genes in A. venetum?

• "The filtered clean reads ranged from 42975126 to 51285576, the total base number was 6150108311 to 7314153091, and the total base number was 6069861180 to 7301018739 after quality control." - There is a repetition of "the total base number." Please clarify this part.

• In the section about differentially expressed genes, you mentioned the p-value being less than 0.05 and a difference multiple greater than 2 as criteria. Could you explain the biological significance of these criteria in identifying differentially expressed genes?

• Can you explain the biological implications of the observed gene expression patterns, such as the upregulation at early stages and downregulation at later stages of stress, in A. venetum under salt stress?

• In the GO classification and functional enrichment analysis, what are the functional categories that the differentially expressed genes are associated with, and how do these categories relate to A. venetum's response to salt stress?

• Could you explain the biological significance of the identified pathways such as "zetin biosynthesis," "phenylpropanoid biosynthesis," and "flavonoid biosynthesis" in the context of A. venetum's response to salt stress?

• When you mention "reverse transporter activity (GO:0015297)," could you clarify what this term means in the context of salt stress and its relationship to salt adaptation?

• What is the significance of the turquoise module containing the most genes and the green-yellow module with the smallest number of genes in the co-expression network analysis?

• How do the identified modules and their correlations with differentially expressed genes relate to the plant's response to salt stress, and what specific functions or processes are associated with each module?

• Can you elaborate on the relationship between the correlation coefficients and the identified modules in the context of A. venetum's adaptation to salt stress?

• In the GO enrichment analysis, how do pathways related to salt stress, such as "water transmembrane transporter activity," "reverse transporter activity," and "response to reactive oxygen species," contribute to A. venetum's salt stress response, and what are the key genes associated with these pathways?

Good Luck!

Reviewer #3: The entire manuscript is well written except material and methods

Need some minor suggested improvements

1. In the Material and Methods section, kindly ensure to explicitly mention the tool or software utilized for the calculation of FPKM in the determination of gene expression values

2. The Material and Methods section appears to be incomplete. We kindly request you to incorporate the following:

• Clearly state the developmental stage of the plant at which the sample was collected for transcriptome analysis

• Provide a detailed introduction to Co-expression network analysis, along with pertinent information regarding related packages such as WGCNA, in the Material and Methods section.

• Elaborate on the techniques and tools employed for the formation of Gene Ontology annotations

3. For enhanced clarity and comprehension, we recommend including a supplementary table presenting the correlation coefficients of gene pairs derived from the WGCNA analysis.

4. Please provide a comprehensive explanation and specification of the twelve modules obtained from the WGCNA analysis.

Reviewer #4: Dear authors

The authors have made an interesting research entitled “Identification of core genes involved in the response of Apocynum venetum to salt stress based on transcriptome sequencing and WGCNA”.

They studied some molecular mechanisms of salt tolerance in A. venetum and introduced some key genes and gene networks that are involved in salt stress regulation and salt tolerance mechanisms in plants.

One concern is about gene IDs. Replace the gene IDs like TRINITY_DN102_c0_g1, TRINITY_DN3073_c0_g1 and TRINITY_DN6732_c0_g1 with GenBank accession numbers or convert those to Arabidopsis orthologues

I think this article is worthy to be published.

Regards,

6. PLOS authors have the option to publish the peer review history of their article (what does this mean?). If published, this will include your full peer review and any attached files.

Reviewer #1: No

Reviewer #2: **Yes: **Dr. Shahzaib Ahamad

Reviewer #3: No

Reviewer #4: **Yes: **ALI MOGHADAM

---

## [Author Response · Author response to Decision Letter 0]

16 Feb 2024

The manuscript has the opinions of four reviewers, and there are many replies. We have written the replies to the reviewers in word and submitted them.

---

## [Decision Letter · Decision Letter 1]

26 Feb 2024

Identification of core genes involved in the response of Apocynum venetum to salt stress based on transcriptome sequencing and WGCNA

PONE-D-23-25332R1

Dear Dr. Zhen,

We’re pleased to inform you that your manuscript has been judged scientifically suitable for publication and will be formally accepted for publication once it meets all outstanding technical requirements.

Kind regards,

Mohammad Sadegh Taghizadeh, Ph.D.

Academic Editor

PLOS ONE

Additional Editor Comments (optional):

Reviewers' comments:

Reviewer's Responses to Questions

**Comments to the Author**

1. If the authors have adequately addressed your comments raised in a previous round of review and you feel that this manuscript is now acceptable for publication, you may indicate that here to bypass the “Comments to the Author” section, enter your conflict of interest statement in the “Confidential to Editor” section, and submit your "Accept" recommendation.

Reviewer #1: All comments have been addressed

Reviewer #3: All comments have been addressed

Reviewer #4: All comments have been addressed

2. Is the manuscript technically sound, and do the data support the conclusions?

Reviewer #1: Yes

Reviewer #3: Yes

Reviewer #4: Yes

3. Has the statistical analysis been performed appropriately and rigorously? 

Reviewer #1: Yes

Reviewer #3: Yes

Reviewer #4: Yes

4. Have the authors made all data underlying the findings in their manuscript fully available?

Reviewer #1: Yes

Reviewer #3: Yes

Reviewer #4: Yes

5. Is the manuscript presented in an intelligible fashion and written in standard English?

Reviewer #1: Yes

Reviewer #3: Yes

Reviewer #4: Yes

6. Review Comments to the Author

Reviewer #1: The authors of this article have made changes to all the review questions, but please pay attention to the use of punctuation.

Reviewer #3: All the comments have been addressed like Discussion part has been improved as per the suggestion and data and results are more meaningful.

Now it is up to to journal's standard

Reviewer #4: Dear authors

They addressed all the comments and now the manuscript is acceptable to be published.

Regards,

7. PLOS authors have the option to publish the peer review history of their article (what does this mean?). If published, this will include your full peer review and any attached files.

Reviewer #1: No

Reviewer #3: **Yes: **Shbana Begam

Reviewer #4: **Yes: **Ali Moghadam
